# Fatal opioid overdoses during and shortly after hospital admissions in England: A case-crossover study

Dan Lewer[1,2,3]*, Brian Eastwood[2], Martin White[2], Thomas D. Brothers[1,3,4], Martin McCusker[5], Caroline Copeland[6,7], Michael Farrell[3], Irene Petersen[8]

1 Department of Epidemiology and Public Health, University College London, London, United Kingdom, 2 Alcohol, Drugs, Tobacco and Justice Division, Public Health England, London, United Kingdom, 3 National Drug and Alcohol Research Centre, University of New South Wales, Kensington, Australia, 4 Department of Medicine, Dalhousie University, Halifax, Canada, 5 Lambeth Service User Council, London, United Kingdom, 6 Institute of Pharmaceutical Sciences, King's College London, London, United Kingdom, 7 National Programme on Substance Abuse Deaths, St George's, University of London, London, United Kingdom, 8 Department of Primary Care and Population Health, University College London, London, United Kingdom

* d.lewer@ucl.ac.uk

**Data Availability Statement:** The data used in this study are held at Public Health England to support programme planning and strategy. Public Health England's information governance protocols and

## Abstract

### Background

Hospital patients who use illicit opioids such as heroin may use drugs during an admission or leave the hospital in order to use drugs. There have been reports of patients found dead from drug poisoning on the hospital premises or shortly after leaving the hospital. This study examines whether hospital admission and discharge are associated with increased risk of opioid-related death.

### Methods and findings

We conducted a case-crossover study of opioid-related deaths in England. Our study included 13,609 deaths between January 1, 2010 and December 31, 2019 among individuals aged 18 to 64. For each death, we sampled 5 control days from the period 730 to 28 days before death. We used data from the national Hospital Episode Statistics database to determine the time proximity of deaths and control days to hospital admissions. We estimated the association between hospital admission and opioid-related death using conditional logistic regression, with a reference category of time neither admitted to the hospital nor within 14 days of discharge. A total of 236/13,609 deaths (1.7%) occurred following drug use while admitted to the hospital. The risk during hospital admissions was similar or lower than periods neither admitted to the hospital nor recently discharged, with odds ratios 1.03 (95% CI 0.87 to 1.21; $p = 0.75$) for the first 14 days of an admission and 0.41 (95% CI 0.30 to 0.56; $p < 0.001$) for days 15 onwards. 1,088/13,609 deaths (8.0%) occurred in the 14 days after discharge. The risk of opioid-related death increased in this period, with odds ratios of 4.39 (95% CI 3.75 to 5.14; $p < 0.001$) on days 1 to 2 after discharge and 2.09 (95% CI 1.92 to 2.28; $p < 0.001$) on days 3 to 14. 11,629/13,609 deaths (85.5%) did not occur close to a hospital admission, and the remaining 656/13,609 deaths (4.8%) occurred in hospital

the ethical approval for this project require that the data are not shared. Researchers can use the linked Hospital Episode Statistics and ONS mortality data used in this article via the NHS Digital Data Request Service, with more information at https://digital.nhs.uk/services/data-access-request-service-dars.

**Funding:** DL is funded by a National Institute of Health Research Doctoral Research Fellowship [DRF-2018-11-ST2-016]. The views expressed are those of the author(s) and not necessarily those of Public Health England, the NHS, the NIHR or the Department of Health and Social Care. TDB is supported by the Dalhousie University Internal Medicine Research Foundation Fellowship, Killam Postgraduate Scholarship, Ross Stewart Smith Memorial Fellowship in Medical Research, and Clinician Investigator Program Graduate Stipend (all from Dalhousie University Faculty of Medicine), a Canadian Institutes of Health Research Fellowship (CIHR-FRN# 171259), and through the Research in Addiction Medicine Scholars (RAMS) Program (National Institutes of Health/National Institute on Drug Abuse; R25DA033211). The funders had no role in study design, data collection and analysis, decision to publish, or preparation of the manuscript.

**Competing interests:** The authors have read the journal's policy and the authors of this manuscript have the following competing interests. MF is Director of the National Drug and Alcohol Research Centre which receives funding from the Australian Federal Government Department of Health. He has also had unrestricted educational grant funding from Indivior, Mundipharma and Seqirius. Other authors declare no competing interests.

**Abbreviations:** ICD-10, International Classification of Diseases-10th Revision; IQR, interquartile range; NHS, National Health Service; STROBE, Strengthening the Reporting of Observational Studies in Epidemiology.

following admission due to drug poisoning. Risk was greater for patients discharged from psychiatric admissions, those who left the hospital against medical advice, and those leaving the hospital after admissions of 7 days or more. The main limitation of the method is that it does not control for time-varying health or drug use within individuals; therefore, hospital admissions coinciding with high-risk periods may in part explain the results.

## Conclusions

Discharge from the hospital is associated with an acute increase in the risk of opioid-related death, and 1 in 14 opioid-related deaths in England happens in the 2 weeks after the hospital discharge. This supports interventions that prevent early discharge and improve linkage with community drug treatment and harm reduction services.

## Author summary

### Why was this study done?

- The number of deaths due to poisoning by opioids such as heroin is increasing in England.
- The risk of dying due to a drug overdose varies across time, for example, deaths are common in the weeks after the release from prison or discharge from drug treatment.
- Hospital patients who use illicit drugs report undertreated pain and opioid withdrawal, and patients have overdosed in hospital toilets and car parks.
- Hospital admission and discharge may be an opportunity to help people who use illicit opioids.

### What did the researchers do and find?

- We studied people who died due to a fatal drug overdose in England, where an opioid such as heroin contributed to the death.
- We looked at the history of hospital admissions for these individuals, and we assessed whether they were admitted to the hospital at the time of death or had recently been discharged.
- We found that fatal opioid overdoses are 4 times more likely in the 2 days after the hospital discharge than at other times, showing that hospital discharge is a high-risk time for people who use illicit opioids.
- We also found that some fatal opioid overdoses happened during hospital admissions, but the number was similar or lower than expected among people who use drugs in the community.

### What do these findings mean?

- People who use illicit drugs such as heroin need extra support when being discharged from the hospital.

- Interventions that reduce the risk of fatal overdose, such as opioid agonist treatment and overdose response training with take-home naloxone (an antidote for opioid overdose), may be beneficial when provided in the hospital.

- Hospitals may need to improve training related to addictions and develop policies to implement these overdose reduction interventions.

## Introduction

People who use illicit opioids such as heroin sometimes report unpleasant experiences when admitted to the hospital for medical treatment. In some cases, hospital staff are suspicious when patients describe their symptoms, believing they are "drug seeking" [1,2]. In other cases, staff are concerned about the safety of giving opioid-based medicines to patients who may be taking opioids from other sources. Sometimes, staff are too busy to verify a patient's usual dose of methadone or buprenorphine or do not have sufficient knowledge or training about opioid dependence [3,4]. These factors can lead to inadequate pain control or delayed or insufficient opioid substitution [4]. Patients have also said that some staff are judgmental about illicit drug use and therefore hide the fact that they use drugs [5].

Opioid withdrawal can lead patients to leave the hospital to buy drugs. Some bring a supply into the hospital to keep them going, and some arrange for dealers to visit them while they are staying on a ward [3]. Surveys of hospital patients who use illicit opioids suggest that in-hospital use is common [5,6]. Using drugs in the hospital is associated with high-risk practices, including using alone in a toilet cubicle, rushing the procedure, taking a bigger dose to reduce the need for top-ups, and not having the usual equipment such a new needle and syringe, a tourniquet, and sterile water [5].

There have been newspaper reports of hospital patients taking heroin and being found dead in a hospital toilet, car park, or another public place close to the hospital [7,8]. However, we do not know how many times this has happened or if hospital admissions increase the risk of opioid-related death. The period after discharge from the hospital may also be risky, because opioid tolerance may be reduced, and patients may be unwell and more susceptible to a drug overdose. This study examines whether the risk of opioid-related death is increased during hospital admission and in the period after discharge. We expected these periods to be associated with increased risk of opioid-related death.

## Methods

### Ethical approval

This study was approved by the Public Health England Research Ethics and Governance Group (PHE REGG), ref R&D412, on October 26, 2020. Data were anonymised before analysis, and personal identifiers such as name, address, or NHS number were not available to the research team.

This is a case-crossover study estimating the risk of opioid-related deaths associated with admission to the hospital. Case-crossover studies measure acute "triggering" effects of transient exposures [9]. They make within-subject comparisons in the exposure status when an event occurred (in this study, when someone died after using opioids) with the exposure status at other times. Case-crossover studies are one of a family of self-controlled study designs that only include participants who experienced an event. These designs focus on the timing of an event, in contrast to traditional epidemiological studies that focus on who experiences an event. This study is addressing the question "do hospital admission and discharge trigger opioid-related deaths?" We chose this design because it allows inclusion of a large proportion of cases and is statistically powerful, and it controls confounding more effectively than a cohort design that might compare people who use opioids and are admitted to the hospital with those not admitted.

## Study participants

We studied opioid-related deaths among people in England aged 18 to 64 between January 1, 2010 and December 31, 2019 based on the date of death (rather than registration). We defined opioid-related deaths as those with an underlying cause of drug poisoning (using the UK Office for National Statistics definition of drug poisoning [10]: the International Classification of Diseases-10th Revision [ICD-10] codes X40-X44, X60-X64, X85, or Y10-14) and where an opioid is also specified (ICD-10 codes T40.0-T40.4 and T40.6) or if opioid dependence (ICD-10 F11) was the underlying cause of death. Coroners investigate drug-related deaths in England, including analysis of toxicology results. This means that the causes of death in this study have been validated to a greater degree than for most deaths. For simplicity, we refer to these deaths as "fatal opioid overdoses" in the title and author summary, although we use "opioid-related deaths" elsewhere to reflect the difficulty of attributing deaths to one specific drug.

Data were drawn from a database that includes mortality data from the UK Office for National Statistics and hospital records from the national Hospital Episode Statistics database, with probabilistic linkage between the 2 sources using the National Health Service (NHS) number, date of birth, sex, and home address [11]. This database does not include deaths if no linkage is found (most likely because the decedent was never admitted to the hospital). Using published mortality data [12], we estimated that 11.8% of deaths were excluded (Fig 1).

## Control days

For each case, we sampled 5 days at random from the period 730 to 28 days prior to death, limiting to the same day of the week as death (Fig 2). The reason for limiting to the same day of the week is that both drug-related deaths and hospital admissions vary by weekday, which may cause confounding. Fewer hospital admissions and discharges occur at the weekend [13], while deaths due to drug poisoning peak on Saturday [14]. We chose the period 730 to 28 days prior to death to avoid control days that are too close to death and may have correlated exposures, while also allowing reasonable exchangeability in the probability of hospital admission. We then observed the exposure status on the control days. In sensitivity analysis, we repeated the study with control days sampled from the periods 365 to 28 days before death and 1,095 to 28 days before death.

## Exposure status

Control days were classified as (A) currently admitted to the hospital (days 1 to 14 after admission); (B) currently admitted to hospital (15+ days after admission); (C) days 1 or 2 after

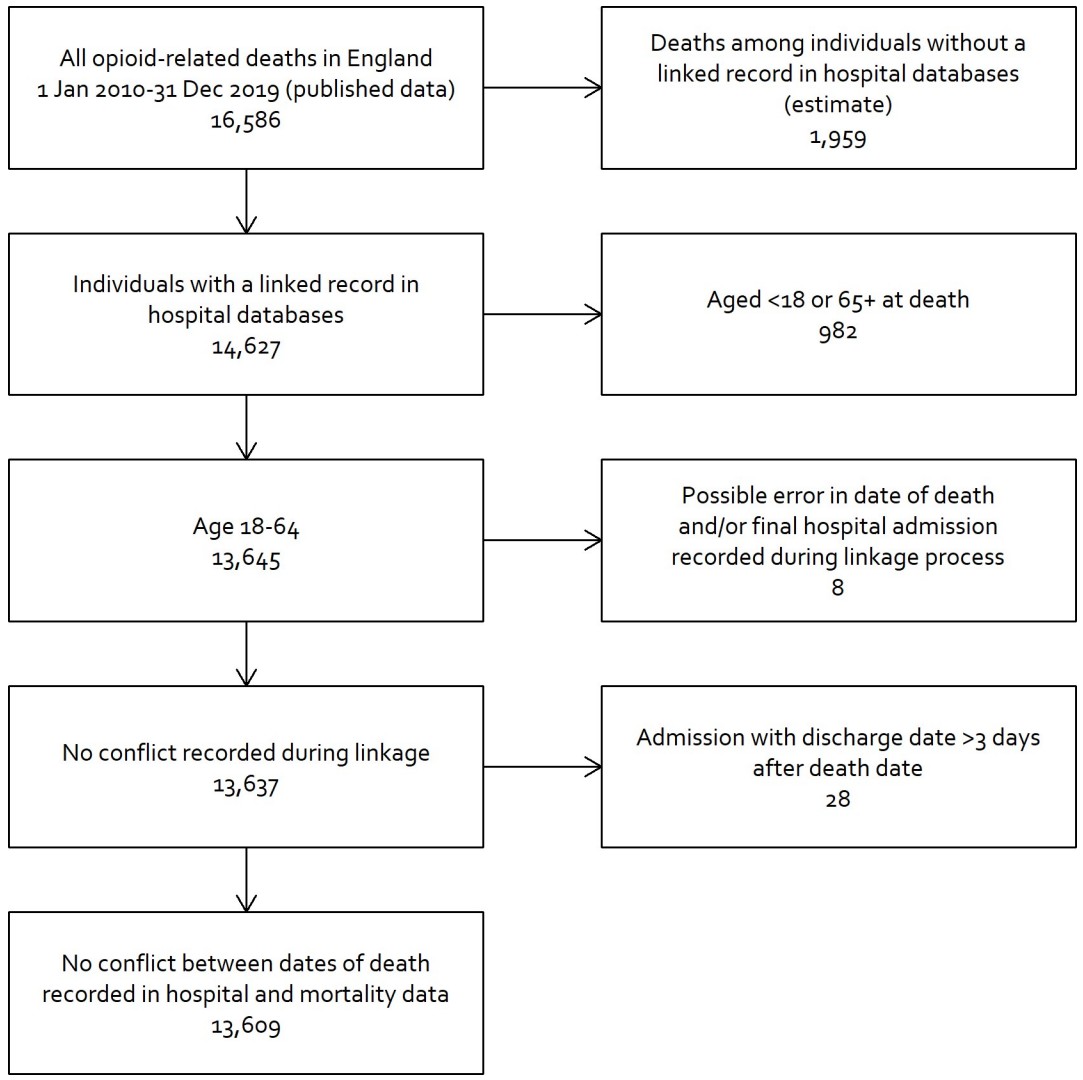

**Fig 1. Derivation of study population.**

discharge; (D) days 3 to 14 after discharge; and (E) neither hospitalised nor recently discharged (i.e., time spent in the community).

To understand additional differences in the risk of opioid-related death, we further classified admissions as psychiatric or nonpsychiatric; whether postdischarge risk periods followed discharge against medical advice or planned discharge; and whether postdischarge periods followed admissions of 1 day, 2 to 6 days, or 7+ days. We classified admissions as "psychiatric" if the patient was admitted to a specialist mental health provider or if the lead treatment specialty was recorded as "mental health." We classified discharges as "against medical advice" if the doctor recorded the discharge method as "self-discharged or discharged by a relative or advocate" [15].

Some deaths occurred in hospital. If a patient was admitted due to drug poisoning (see Fig A in S1 Appendix for detail on how we classified these admissions), we moved the date of death to immediately prior to admission and censored the final admission. If a patient died on the day of discharge and hospital data showed that the patient died in the hospital, the death

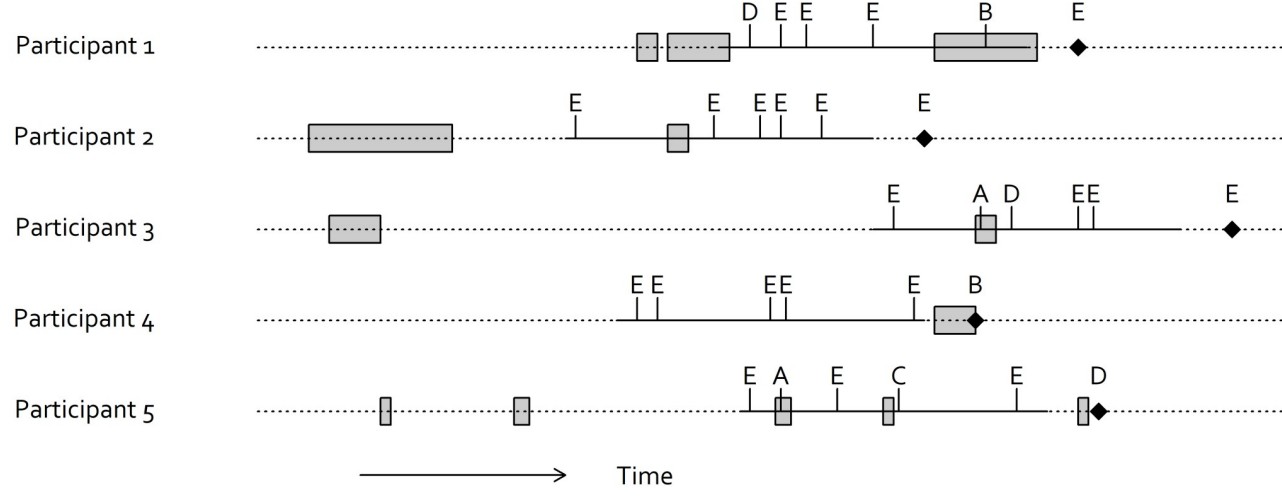

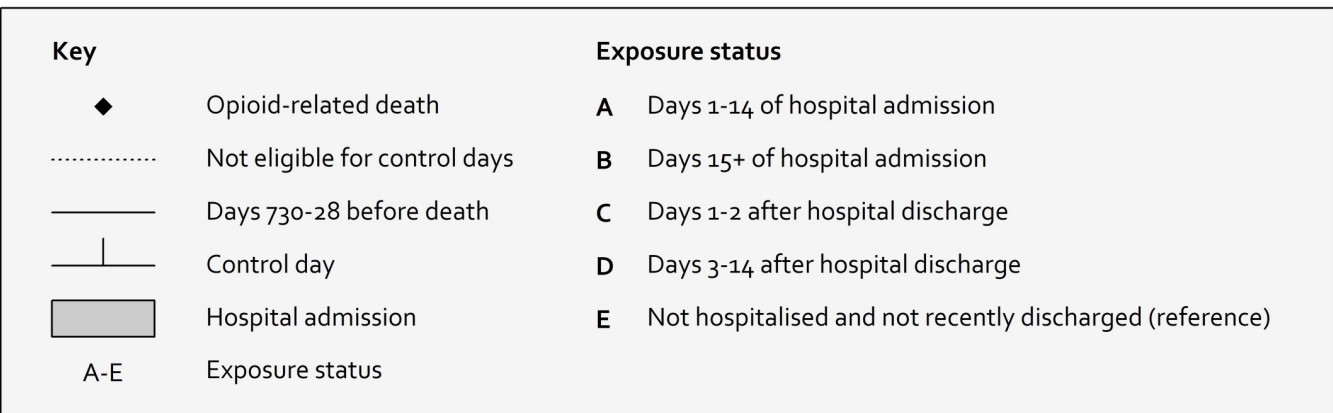

**Fig 2. Illustration of patient timelines and selection of control windows for 5 participants.** Exposure status at time of opioid-related death is compared with 5 days sampled between 730 and 28 days before death.

was assigned to exposure status A or B (currently in hospital), while if the hospital data showed that the patient was discharged alive, exposure status C was assigned (discharged in the past 1 to 2 days).

For 28 individuals, we found discordant data in which hospital discharge dates were recorded long after death dates. These discrepant dates are likely due to failed linkage rather than inaccuracies in hospital discharge or death dates, and we excluded these individuals from the study. A small number of individuals had admissions in the 3 days after death, for example, for organ donation, and these cases were retained.

## Statistical analysis

We described the characteristics of deaths included in the study and then used conditional logistic regression (stratified by individual) to estimate the association between hospital admission and opioid-related death. The reference category was time neither admitted to hospital nor recently discharged (category E). We repeated the analysis with stratification by sex. We published a protocol before doing analysis [16]. In our protocol, the primary analysis was a self-controlled case series design and have included results using this approach in Fig D and

Table B in S1 Appendix. We chose a case-crossover design instead because it allowed analysis of exposures with varying durations. Analysis was conducted using R version 4.0.3.

This study is reported as per the Strengthening the Reporting of Observational Studies in Epidemiology (STROBE) guideline [17] (see Table D in S1 Appendix).

## Results

### Description of cases

The study included 13,609 opioid-related deaths. A total of 9,765/13,609 decedents (71.8%) were male, 12,437/13,609 (91.4%) had "white" ethnicity, the median age at death was 42 (interquartile range [IQR] 35 to 49), and decedents predominantly lived in deprived neighbourhoods. Table 1 summarises the characteristics of cases.

Most opioid-related deaths (11,629/13,609, 85.5%) did not occur in the hospital or within 14 days of discharge. A total of 656/13,609 deaths (4.8%) occurred in hospital following admission due to drug poisoning. 236/13,609 deaths (1.7%) occurred during a hospital admission where the patient was admitted for a reason other than drug poisoning. The remaining 1,088/13,609 deaths (8.0%) occurred in the 14 days after discharge. Based on published data, we estimated that our study excluded 11.8% of deaths because the decedent was never admitted to the hospital (Fig 1). This suggests that in the population, 7.1% of opioid-related deaths occur in the 14 days after discharge.

### Characteristics of hospital admissions in the 2 years prior to death

A total of 37,570 hospital admissions occurred in the 730 days prior to death, with a median of 1 (IQR 0 to 4) per individual. 3,742/37,570 (10.0%) of admissions ended in discharge against medical advice, and 3,418/37,570 (9.1%) were classified as psychiatric admissions. Table 2 summarises the characteristics of hospital admissions in the 2 years prior to death.

### Results of case-crossover analysis

Days 1 to 14 of hospital admissions had a similar risk of opioid-related death as periods in the community (conditional odds ratio of 1.03; 95% CI 0.87 to 1.21; $p = 0.95$). Days 15+ of hospital admission were associated with lower risk of opioid-related death (conditional odds ratio of 0.41; 95% CI 0.30 to 0.56; $p < 0.001$).

The risk of opioid-related death increased substantially after discharge, with a conditional odds ratio of 4.39 (95% CI 3.75 to 5.14; $p < 0.001$) in days 1 to 2 after discharge and 2.09 (95% CI 1.92 to 2.28; $p < 0.001$) in days 3 to 14. The risk was higher for people discharged after a psychiatric admission and for people who left the hospital against medical advice. Longer admissions were associated with greater risk of opioid-related death after discharge, and we observed this gradient in days 1 to 2 after discharge and days 3 to 14. Results of the case-crossover analysis are shown in Fig 3. Sex-stratified results suggested similar associations for men and women. Sensitivity analysis with control days sampled from 365 to 28 days before death showed slightly smaller associations between hospital discharge and opioid-related death, and sensitivity analysis with control days sampled from 1,095 to 28 days before death showed slightly larger associations. Full results for stratified analyses and sensitivity analyses are provided in Figs D and E and Table C in S1 Appendix.

## Discussion

In this nationwide study of opioid-related deaths in England over 10 years, we found that the risk of death is very high in the 2 weeks after hospital discharge, and this period accounts for 1

**Table 1. Characteristics of individuals included in the study.**

| Variable | Level | Number (%) |
|---|---|---|
| Total | | 13,609 (100.0) |
| Age at death | Median [IQR] | 42 [35–49] |
| | Mean [SD] | 41.9 [10.0] |
| Sex | Male | 9,765 (71.8) |
| | Female | 3,844 (28.2) |
| Ethnicity* | White British, White Irish, or Other White | 12,437 (91.4) |
| | Asian or Asian British | 193 (1.4) |
| | Other | 145 (1.1) |
| | Black or Black British | 133 (1.0) |
| | Mixed | 99 (0.7) |
| | Unknown | 602 (4.4) |
| Deprivation (IMD) | 1: Most deprived | 6,067 (44.6) |
| | 2 | 3,370 (24.8) |
| | 3 | 1,940 (14.3) |
| | 4 | 1,328 (9.8) |
| | 5: Least deprived | 790 (5.8) |
| | Missing | 114 (0.8) |
| Year of death | 2010–2011 | 2,127 (15.6) |
| | 2012–2013 | 2,442 (17.9) |
| | 2014–2015 | 3,094 (22.7) |
| | 2016–2017 | 3,323 (24.4) |
| | 2018–2019 | 2,623 (19.3) |
| Geographical region | North West | 2,550 (18.7) |
| | South East | 1,894 (13.9) |
| | Yorkshire and The Humber | 1,636 (12.0) |
| | South West | 1,384 (10.2) |
| | West Midlands | 1,356 (10.0) |
| | London | 1,236 (9.1) |
| | East of England | 1,184 (8.7) |
| | North East | 1,112 (8.2) |
| | East Midlands | 798 (5.9) |
| | Missing | 459 (3.4) |
| Proximity in time to hospital admission | Died in hospital (admitted after opioid use) | 656 (4.8) |
| | Died in hospital (admitted for other reasons) | 236 (1.7) |
| | Died in the 14 days after discharge | 1,088 (8.0) |
| | Not in hospital or within 2 weeks of discharge | 11,629 (85.5) |

* Ethnicity is derived from the hospital data. Where a participant had hospital admissions with different recorded ethnic categories, we used the most commonly recorded category or the most recent category where multiple categories had the same frequency.
IMD, Index of Multiple Deprivation, derived from the Lower Super Output Area of the patient's home address; IQR, interquartile range.

in 14 deaths. Patients who leave the hospital against medical advice and those leaving the hospital after a longer stay have higher risk. We also identified 236 cases where the data suggest that an opioid-related death occurred during a hospital admission. While each of these cases is a serious and potentially preventable incident, the overall results of our study suggest that the

**Table 2. Characteristics of hospital admissions in the 2 years prior to opioid-related deaths in England from January 1, 2010 to December 31, 2019.**

| Variable | Level | Number (%) |
|---|---|---|
| Total | | 37,570 (100.0) |
| Discharged AMA | Yes | 3,742 (10.0) |
| | No | 33,828 (90.0) |
| Length of admission (days) | 1 | 15,026 (40.0) |
| | 2–6 | 14,615 (38.9) |
| | 7+ | 7,929 (21.1) |
| | Median [IQR] | 2 [1–5] |
| | Mean [SD] | 8.7 [38.2] |
| Psychiatric admission | No | 34,152 (90.9) |
| | Yes | 3,418 (9.1) |
| Drug poisoning | No | 32,625 (86.8) |
| | Yes | 4,945 (13.2) |

AMA, discharge against medical advice; IQR, interquartile range.

risk of opioid-related death during hospital admissions is similar or lower than during time spent in the community.

## Comparison with other studies

We are aware of one other study that has investigated deaths due to drug poisoning in relation to hospital admissions [18,19]. This study used a cohort of people registered for drug treatment in Scotland and reported the rate of drug-related deaths according to time proximity to hospital admissions. The referent in this study was the mortality rate among people who were never admitted to the hospital, and the results may therefore reflect differences between people who were admitted to the hospital and those who were not. The associations are much greater than in our study; for example, the mortality rate in the 28 days after discharge is 15 times that of individuals who were never admitted. Even periods more than 1 year after discharge have 3 times the rate, suggesting that these differences are unlikely to relate to the hospital admission itself.

Research has shown that other life events are also associated with opioid-related deaths. In particular, studies in several countries have found a high risk of drug-related death immediately after the release from prison [20–23] and a protective effect of opioid agonist therapy during this period [24]. Another example is the first 2 weeks after cessation of community-based opioid agonist therapy [25]. These are times when opioid agonist therapy is interrupted, tolerance is changing, and people may get drugs from a different source or use drugs in different ways.

## Strengths and limitations

By studying all opioid-related deaths in England, we were able to include people who have never been in drug treatment, a group that is often excluded from studies of this population. The design also meant that we were able to estimate the absolute number of opioid-related deaths that occurred during hospital admissions (236 over 10 years). It also provided power to observe the risk associated with different types and durations of hospital admission, which would be challenging even with an extremely large cohort study.

While the self-controlled methodology eliminates time-invariant confounders, the results may partially be explained by escalating drug use or deteriorating health over time. It is

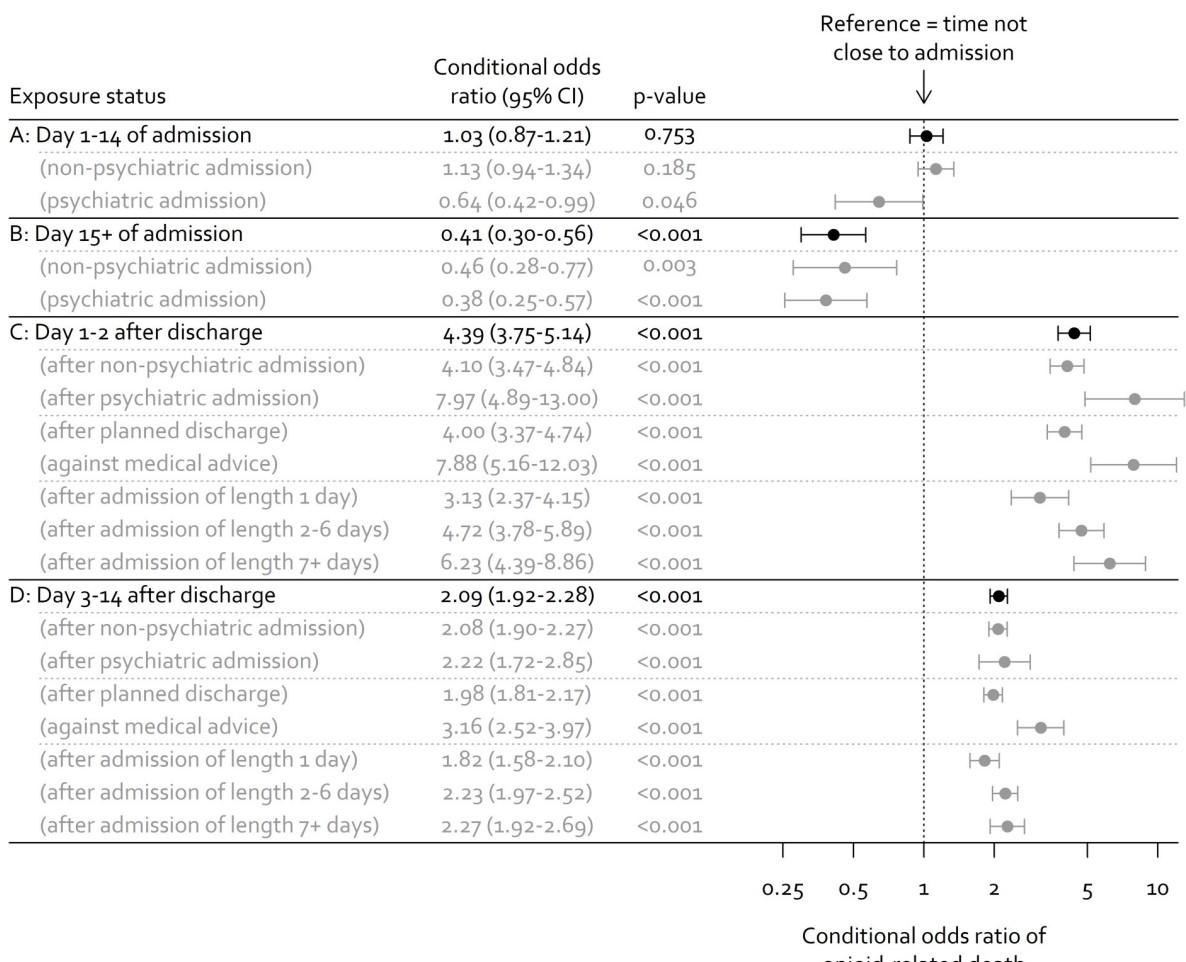

| Exposure status | Conditional odds ratio (95% CI) | p-value |
|---|---|---|
| A: Day 1–14 of admission | 1.03 (0.87–1.21) | 0.753 |
| (non-psychiatric admission) | 1.13 (0.94–1.34) | 0.185 |
| (psychiatric admission) | 0.64 (0.42–0.99) | 0.046 |
| B: Day 15+ of admission | 0.41 (0.30–0.56) | <0.001 |
| (non-psychiatric admission) | 0.46 (0.28–0.77) | 0.003 |
| (psychiatric admission) | 0.38 (0.25–0.57) | <0.001 |
| C: Day 1–2 after discharge | 4.39 (3.75–5.14) | <0.001 |
| (after non-psychiatric admission) | 4.10 (3.47–4.84) | <0.001 |
| (after psychiatric admission) | 7.97 (4.89–13.00) | <0.001 |
| (after planned discharge) | 4.00 (3.37–4.74) | <0.001 |
| (against medical advice) | 7.88 (5.16–12.03) | <0.001 |
| (after admission of length 1 day) | 3.13 (2.37–4.15) | <0.001 |
| (after admission of length 2–6 days) | 4.72 (3.78–5.89) | <0.001 |
| (after admission of length 7+ days) | 6.23 (4.39–8.86) | <0.001 |
| D: Day 3–14 after discharge | 2.09 (1.92–2.28) | <0.001 |
| (after non-psychiatric admission) | 2.08 (1.90–2.27) | <0.001 |
| (after psychiatric admission) | 2.22 (1.72–2.85) | <0.001 |
| (after planned discharge) | 1.98 (1.81–2.17) | <0.001 |
| (against medical advice) | 3.16 (2.52–3.97) | <0.001 |
| (after admission of length 1 day) | 1.82 (1.58–2.10) | <0.001 |
| (after admission of length 2–6 days) | 2.23 (1.97–2.52) | <0.001 |
| (after admission of length 7+ days) | 2.27 (1.92–2.69) | <0.001 |

**Fig 3. Risk of opioid-related death according to time proximity to hospital admission (results of case-crossover analysis).**

possible, for example, that someone in poor health or at times of more intense drug use would be more likely to be admitted to the hospital and more susceptible to death after using opioids. We tried to limit this type of confounding by selecting control days no more than 2 years before death. The density of hospital admissions does increase before death, but this increase is gradual (Fig C in S1 Appendix).

Our data did not include detailed information about drug use or enrollment in drug treatment services. We therefore could not confirm that participants were using drugs on the control days. People who used illicit opioids during the past decade in England have mostly been using drugs for many years. For example, in a cross-sectional survey of people who injected drugs in England in 2019, the median duration of drug use was 16 years, and only 6% of participants had injected for less than 2 years [26].

Our use of a national hospital dataset meant that we were able to include all hospital admissions in England, but it also meant that we had limited detail about individual patients. One limitation relates to "discharge against medical advice." This was a binary variable in our analysis, when in reality there are a range of scenarios where discharge is negotiated between patients and hospital staff. A second limitation relates to our classification of hospital admissions ending in opioid-related death. If an admission ended in opioid-related death and the

primary cause of admission was a common complication of opioid poisoning, such as cardiac arrest, then we assumed that the indication for admission was related to opioid poisoning. This is a sensitive approach, and, in some cases, the opioid poisoning may have happened while the patient was in the hospital. Therefore, it is possible that we have underestimated the association between hospital admissions (exposure periods A and B) and opioid-related death.

Although our definition of opioid-related death is widely used, its validity is not known. It can be difficult to determine the cause of death when someone dies suddenly and alone. It is possible that some participants in our study who died shortly after a hospital admission died for reasons related to the admission (such as an acute infection) rather than due to opioid poisoning. If the cause of death is unclear but the individual was known to use illicit drugs, a doctor may assume that the death was primarily due to drug use. This type of misclassification may partly explain the association between hospital discharge and opioid-related death.

## Interpretation

We identified 3 reasons why discharge from the hospital may be associated with increased risk of opioid-related death. First, opioid tolerance could reduce during an admission. Animal models suggest that opioid tolerance has a half-life of 6 days [27], supporting reductions in tolerance after admissions of 2 to 6 days or 7+ days (the categories used in our analysis). We also saw increased risk after admissions of only 1 day, suggesting that other mechanisms are also important. Second, opioid agonist therapy may be interrupted or reduced either at admission or discharge (see Box 1). This may further contribute to reduced tolerance and increase the likelihood that patients will use riskier opioids such as injected heroin. Third, patients may have an acute illness that makes them more vulnerable to death after using opioids, particularly respiratory problems such as pneumonias and acute exacerbations of chronic obstructive pulmonary disease. These are common reasons for hospital admission in this population [28] and increase the risk associated with central nervous system depressants. Painful conditions may also be important because they are associated with increased use of illicit and prescribed opioids.

### Box 1. Interpretation by a client representative at a community drug and alcohol service

I know from personal experience and that of my peers that hospitals can be hostile, particularly when you are admitted in an emergency. Planned stays give you time to get prepared and make sure you have enough drugs to carry you through. When the stay is unplanned, you are reliant on the doctors giving you methadone or buprenorphine. They are not experts in this field and can be suspicious or at best conservative with their doses. Although some staff do their best to help, it is often made clear they suspect you are "drug seeking." You have to beg to get the help you need. Many of my peers have left the hospital early in withdrawal and pain. They might be buying drugs from someone they do not know, perhaps unwell and with reduced tolerance, then using in an alley or public toilet. It is not surprising that so many people die due to drug overdoses in the days after leaving the hospital. Part of the solution is better communication between the local drug services and the hospital. This could help patients get the medication they need to stay in the hospital and help people arrange transport, accommodation, and timely opioid substitution when they leave.

We observed a high risk soon after hospital discharge and estimate that 1 in 14 opioid-related deaths in the population occur during these 2 weeks. The safety of discharge may be improved with better linkage to community drug treatment services, and there is a need for research into interventions that can improve continuity of opioid agonist therapy between community and hospital settings. It is thought that half of people who die after using illicit opioids in England have never been in contact with drug treatment services [29]. Initiation of opioid agonist therapy in the hospital is therefore also important. A randomised trial showed that patients that started on buprenorphine in hospital and referred to a community drug service are more likely to continue with treatment and less likely to use illicit opioids, compared with those assigned to opioid detoxification [30]. Hospitals can also provide advice on reducing overdose risk, such as using small test doses and not using drugs alone, and tools to help reduce risk, such as basic life support training and take-home naloxone [31].

We found variation in the risk of opioid-related death in our detailed exposure periods. In particular, the period after discharge against medical advice was associated with opioid-related death. Discharge against medical advice may happen when a patient is experiencing pain or withdrawal and leaves the hospital to use illicit opioids. The finding that the risk of opioid-related death reduces after day 14 of a hospital admission should be treated with caution, because the subset of admissions that are longer than 14 days may be unusual. For example, these patients may have less severe drug dependence or better controlled pain and be less at risk of opioid-related death for these reasons.

In many countries, the average age of people who use illicit opioids is increasing, and the frequency of long-term conditions is also increasing [32]. People who use illicit opioids do not always seek timely healthcare, in part due to fear of stigma, opioid withdrawal in the hospital, and poor pain management [33]. Hospitals that enable patients to disclose illicit drug use without fear of discrimination will be a central element of accessible and high-quality hospital care for this population.

## Conclusions

Discharge from the hospital is associated with an acute increase in the risk of opioid-related death, and 1 in 14 opioid-related deaths in England happens in the 2 weeks after the hospital discharge. This supports interventions that prevent early discharge and improve linkage with community drug treatment and harm reduction services.

## Supporting information

**S1 Appendix. Table A:** Distribution of ICD-10 diagnoses in opioid-related deaths in England between January 1, 2010 and December 31, 2019. **Table B:** Results of alternative self-controlled methodologies. Values are conditional odds ratios of opioid-related deaths (95% CIs). **Table C:** Results of case-crossover analysis stratified by sex and calendar year of death. Values are conditional odds ratio of opioid-related death (95% CIs). **Table D:** STROBE Checklist. **Fig A:** Flowchart showing how the exposure status on the day of death was determined. **Fig B:** Distribution of age at death for 13,609 people who died due to fatal opioid overdose in England between January 1, 2010 and December 31, 2019. Deaths at ages under 18 or over 65 were excluded from the study. **Fig C:** Number of hospital admissions in the 730 days prior to death among 13,609 people who died due to opioid overdose in England between January 1, 2010 and December 31, 2019, by 10-day period. **Fig D:** Results of alternative self-controlled methodologies. The chart shows conditional odds ratios with 95% CIs. **Fig E:** Results of case-crossover analysis stratified by sex and calendar year of death. Values are conditional odds ratio of opioid-related death (95% CIs). ICD-10, International Classification of Diseases-10th Revision;

STROBE, Strengthening the Reporting of Observational Studies in Epidemiology.
(PDF)

## Author Contributions

**Conceptualization:** Dan Lewer, Brian Eastwood, Martin White, Martin McCusker, Caroline Copeland, Michael Farrell, Irene Petersen.

**Data curation:** Dan Lewer.

**Formal analysis:** Dan Lewer.

**Funding acquisition:** Dan Lewer.

**Investigation:** Dan Lewer, Brian Eastwood, Martin White, Thomas D. Brothers, Martin McCusker, Caroline Copeland, Michael Farrell, Irene Petersen.

**Methodology:** Dan Lewer, Brian Eastwood, Martin White, Thomas D. Brothers, Irene Petersen.

**Software:** Dan Lewer.

**Supervision:** Brian Eastwood, Martin White, Martin McCusker, Irene Petersen.

**Visualization:** Dan Lewer.

**Writing – original draft:** Dan Lewer.

**Writing – review & editing:** Dan Lewer, Brian Eastwood, Martin White, Thomas D. Brothers, Martin McCusker, Caroline Copeland, Michael Farrell, Irene Petersen.

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
