## [Editor Report · Decision Letter 0]

15 Mar 2021

Dear Dr Lewer, 

Thank you for submitting your manuscript entitled "Opioid-related deaths during and shortly after hospital admissions in England: case-crossover study" for consideration by PLOS Medicine.

Your manuscript has now been evaluated by the PLOS Medicine editorial staff and I am writing to let you know that we would like to send your submission out for external peer review.

Please re-submit your manuscript within two working days, i.e. by March 18, 2021.

Kind regards,

Beryne Odeny

Associate Editor

PLOS Medicine

---

## [Decision Letter · Decision Letter 1]

30 Apr 2021

Dear Dr. Lewer,

Thank you very much for submitting your manuscript "Opioid-related deaths during and shortly after hospital admissions in England: case-crossover study" (PMEDICINE-D-21-01225R1) for consideration at PLOS Medicine. 

[LINK]

In light of these reviews, I am afraid that we will not be able to accept the manuscript for publication in the journal in its current form, but we would like to consider a revised version that addresses the reviewers' and editors' comments. Obviously we cannot make any decision about publication until we have seen the revised manuscript and your response, and we plan to seek re-review by one or more of the reviewers. 

We expect to receive your revised manuscript by May 21 2021 11:59PM. Please email us (plosmedicine@plos.org) if you have any questions or concerns.

We look forward to receiving your revised manuscript. 

Sincerely,

Beryne Odeny, 

PLOS Medicine 

plosmedicine.org

• Abstract summary - At this stage, we ask that you reformat your non-technical Author Summary. The Author Summary should immediately follow the Abstract in your revised manuscript. This text is subject to editorial change and should be distinct from the scientific abstract. The summary should be accessible to a wide audience that includes both scientists and non-scientists. Please see our author guidelines for more information: https://journals.plos.org/plosmedicine/s/revising-your-manuscript#loc-author-summary.

• Abstract:

1. Please structure your abstract using the PLOS Medicine headings (Background, Methods and Findings, Conclusions). Please combine the Methods and Findings sections into one section, “Methods and findings”.

2. Please ensure that all numbers presented in the abstract are present and identical to numbers presented in the main manuscript text.

3. Please include the actual amounts or percentages of relevant outcomes, not just hazard ratios.

4. Please include the important dependent variables that are adjusted for in the analyses.

5. Please quantify the main results (with p values in addition to 95% CI).

6. In the last sentence of the Abstract Methods and Findings section, please describe the main limitation(s) of the study's methodology.

• Please ensure that the study is reported according to the STROBE guideline, and include the completed STROBE checklist as Supporting Information. Please add the following statement, or similar, to the Methods: "This study is reported as per the Strengthening the Reporting of Observational Studies in Epidemiology (STROBE) guideline (S1 Checklist)." The STROBE guideline can be found here: http://www.equator-network.org/reporting-guidelines/strobe/

• In the ethics statement in the manuscript, please provide additional information about the patient records used in your retrospective study. Specifically, please ensure that you have discussed whether all data were fully anonymized before you accessed them and/or whether the IRB or ethics committee waived the requirement for informed consent.

• Your study is observational and therefore causality cannot be inferred. Please remove language that implies causality, such as “risk of” or “risk”. Refer to associations instead.

• Please include p-values in tables and main text. Please specify the statistical test used to derive the p values.

• Please use the "Vancouver" style for reference formatting and see our website for other reference guidelines https://journals.plos.org/plosmedicine/s/submission-guidelines#loc-references.

Comments from the reviewers:

Reviewer #1: Thanks for the opportunity to review this paper, which uses large nationally representative datasets and a thoughtful approach to answer a critical public health question. I think these results should be published in a high impact open access journal, such as PLOS Medicine, but I think they could be presented in a clearer way, and I think additional limitations should be included in the discussion.

Note, I am no expert on case only designs, nor am I a statistician. I comment as an epidemiologist and as a clinician whose practice includes caring for hospitalised patients with a history of injecting drug use. Peer review by a statistician or an expert in case only methods would be valuable. 

MAJOR POINTS

1. To my mind, the central issue here is to what extent the ICD-10 codes selected accurately identify deaths from opiate overdose, which is the focus of the paper.

It would be nice to see some discussion of the codes chosen in the main text of the manuscript - e.g. why include codes relating to Parkinson's medications or anti rheumatic drugs? It would also be nice if the authors could include a breakdown of the codes assigned for the patients in the study - e.g. if the vast majority were X42 or Y42, I would find that more persuasive.

An alternative explanation for the pattern of deaths observed would be that they relate directly to the hospital admission - e.g. partially treated infection, mental health crisis, venous thromboembolism - but that when a young person with a history of opiate dependence is found dead in the community, perhaps surrounded by injecting paraphernalia, doctors completing the death certificate assume that the cause of death is an overdose. This alternative explanation would be consistent with the higher mortality seen following discharges against medical advice and, possibly, with the high mortality seen following longer admissions (presumably these were associated with more severe underlying illness).

It would be nice to include a detailed discussion of this issue, as it is key to the conclusions we draw from the results. Do we know what mortality in the post discharge period looks like in people without a history of opiate dependence, particularly following admission with similar problems (bacteraemias, liver failure, schizophrenia, etc)? 

2. It would be good to include greater justification of some of the analytical choices made - e.g. why only look at 5 control days? Why 14 days post discharge as the period of interest? 

3. The issues regards data availability are problematic but, I suspect, outside the authors' control. 

4. In the introduction, it would be good to include some justification for focusing on 'opioid related deaths' as opposed to 'opiate overdoses'. I note these terms are used slightly interchangeably in the manuscript. I would also like to see some statement about the frequency of these deaths in the population.

5. Inclusion in the study was conditional on having ever been admitted to hospital. It is good that the number of deaths excluded are quantified. It would be good to include some discussion of whether this may result in bias. I cannot immediately see that this would set up, e.g., collider bias. And I am not sure, anyway, whether that would be an issue in a case only analysis?

6. The authors have deviated significantly from the analysis planned in the published protocol. The discussion of the reasons for this in supplementary materials is good. However, this deviation from the published protocol should be mentioned more prominently in the main text of the manuscript. Will the analysis of the NPSAD data be presented separately?

7. There is good discussion about discrepant documentation of dates of discharge and death. However, these discordant dates do raise questions about the accuracy of the dates coded in the two databases - errors won't always be so obvious. Do we know anything about how accurate these data are? If so, some comment on this would be valuable, given accurate dates are key to the analysis presented.

8. A clear statement about whether any attempt was made to adjust for time varying confounders is needed.

9. I think 'Died in hospital following admission with an overdose' (or similar) (as in the tables) is clearer than 'occurred in hospital following drug use in the community' (as used in the text). It seems likely that many/most of the cohort were using drugs in the community.

10. Do we know anything about how accurately discharge against medical advice is coded in HES? Note, this variable is not always binary - sometimes clinicians try to accommodate patients' preferences for a shorter admission, where they would have preferred a longer one. Such instances won't be coded as 'discharge against medical advice'.

11. The paragraph at the end of page 9, which finishes on page 10, is not clearly written. An attempt is made to describe, in a single sentence, the reasons for hospital admission both overall, and in those who died following overdose in hospital. The sentence ends up being very long, with lots of semi colons. This should be rewritten.

12. The relative risk of death during days 1-14 of hospital admission in the abstract differs to the relative risk presented in the results section. I would give the figures for the relative risk of fatal overdose on days 15+ of hospital admission in the main text of the manuscript (results section). 

13. I think the relative protection from overdose provided by a psychiatric admission, or by a medical admission of longer duration, is worth commenting on.

14. Paragraph at the end of page 12 - this is good. It might be worth explicitly discussing pain here. Many conditions that warrant admission to hospital are painful, and therefore the association between overdose and admission to hospital may be driven by increased opiate consumption (both illicit and prescribed) to manage associated pain. Pain could also be discussed in the relevant paragraph on page 13 (immediately after the 'Interpretation' header). 

15. The protocol describes a planned analysis looking at risk of death during the 14 days prior to hospital admission ('period Z'). I assume this was not done because it is not possible - you cannot know whether people who die are in period Z, because people are not admitted to hospital post mortem. Some comment about this planned analysis would be valuable. The planned disaggregation of the overdose deaths into accidental and intentional, which I think is possible, would be good to see.

16. The limitations section in the protocol is better than the limitations section in the manuscript - much of the discussion from the protocol should be included in the manuscript.

17. The 'client representative' comment is excellent. 

18. The figure on page 6 of supplementary materials is unreadable. I think this should be split into a number of separate figures. Note, the reference to this figure in the text on the previous page is missing a number. 

MINOR POINTS

1. Only two authors seem to have made financial disclosures. Who paid for this work? If the other authors have no disclosures, that should be stated. 

2. When the manuscript is resubmitted, please include line numbers, as this makes things easier for reviewers!

3. In Figure 2, it would be good to define the dotted lines in the legend. I would also say 'control day', rather than 'control period'.

4. To my mind, the opposite of acute is elective, and the opposite of psychiatric is non-psychiatric. I suggest 'non-psychiatric' would be a better term than 'acute', given there might have been some planned admissions.

5. In the first paragraph after the 'Interpretation' header, should it read 'more vulnerable to death following an opioid overdose'?

6. Is it worth discussing patient and peer BLS training, in addition to naloxone provision, as potential interventions to prevent death from overdose?

7. I agree with the final paragraph, but it seems an odd way to conclude the manuscript. 

Dr Tom Yates

Imperial College London

Reviewer #2: Thanks for the opportunity to review your manuscript. My role is as a statistical reviewer - my comments and queries focus on the data and analysis presented. I have put general comments and questions first and followed these with queries specific to a section of the manuscript (with a Page reference).

This is clearly written and succinct manuscript that examines an interesting question. There is a protocol available and the analyses carried out match the description provided there. Tables and figures are clear and there is useful supplementary material available. 

There is a planned sensitivity analysis using several different approaches to dealing with dependence of death with exposure period, and the effect of different control period. These results are robust to the different approaches to the death-exposure period issue. There effect estimate is shifted away from the null with a shorter period for the control windows. I don't think this is a problem - at the more reasonable period limitations it is similar to the main results, but I was interested in your interpretation as to the association between shorter periods from control windows and a more 'extreme' effect estimate.

P2. Findings - 1.7%, not clear what the denominator is (of total deaths, or total exposures)

The 'similar or lower' should be qualified to the different periods - a bit confusing otherwise.

P4. Study participants. Do the external cause and ICD-10 codes distinguish between iatrogenic opioid poisonings and others? Presumably medication error could lead to an adverse drug event (particularly in someone vulnerable e.g. with poor kidney or hepatic function). 

Is the external cause data available specific to agents (e.g. prescription vs. non-prescription opioid forms)?

There is some discussion of this in the protocol but not in the main manuscript or supplementary appendix

P5. Was there any evidence for time-trends, i.e. overall changes in opioid deaths in hospitals? If there were this would presumably bias the results towards a positive association.

P6. Figure 2. Nice figure - my only quibble is that the diamond marker can overlap into a hospital admission period on the figure even where it occurs outside the period. Perhaps a line or other marker instead?

Reviewer #3: PLOS Medicine

Re. Manuscript Number: PMEDICINE-D-21-01225R1

Title: How prescription drug monitoring programs influence clinical practice: A mixed

methods systematic review and meta-analysis

Overall: 

This well-written article by Lewer et al. assessed whether patients had a higher risk of opioid-related death after admission for inpatient treatment for causes other than opioid use. The analysis covered ten years of mortality data in the UK. The main findings were: 1) inpatients had no higher odds of an opioid-related death while admitted than people in the community; and 2) patients had higher odds of an opioid-related death after discharge (1-2 days and 3-14 days) when compared to the risk of dying while in the community. I commend the authors for addressing such an important topic. I do not have major concerns, as I believe that the rationale for the study was well described, the description of the methodology was straightforward and easy to understand, the results and conclusions answered the central question of interest stated in the study's objective. I do have minor suggestions:

- Introduction: add references for statements "Sometimes staff are too busy to verify a patient's usual dose of methadone or buprenorphine, or do not have sufficient knowledge or training about opioid dependence" and "Opioid withdrawal can lead patients to leave hospital to buy drugs. Some bring a supply into hospital to keep them going, and some arrange for dealers to visit them while they are staying on a ward."

- Methods, exposure status: Consider adding clear language stating that the sample included patients who had an opioid-related death but who were admitted for causes other than opioid use.

- Methods, statistical analysis: Consider adding more information about the approach used for the analysis, why it was selected as the best tool to analyze the data.

- Discussion: results by gender have some interesting findings (currently in the appendix); consider including this stratified result by gender in the results and discussion.

Reviewer #4: In this case-crossover analysis, the authors investigate whether the period during and just after hospitalization is a time of heightened risk for opioid-related death. They find that the risk of opioid-related death increased substantially after discharge, with the highest risk in days 1-2 after discharge. The risk was higher for people discharged after a psychiatric admission, people who left hospital against medical advice, and for those with longer admissions. These findings have good physiologic basis. The chosen approach (case-crossover) is an appropriate and robust way to answer the study question, which minimizes confounding by time-invariant confounders, since patients are compared to themselves. The authors appropriately note that their findings may be partially be explained by escalating drug use or deteriorating health over time. This is a well-written manuscript with interesting findings. I have just a few comments/suggestions:

Major issues:

Confounding by escalating drug use/deteriorating health over time: The authors appropriately note that their findings may be partially be explained by escalating drug use or deteriorating health over time. This is a strong limitation. I wonder if they could do a sensitivity analysis extending the control window back just 1 year instead of 2, and see if their results are similar (is this what was done in appendix 4? It's not totally clear to me. If so, would move these results regarding sensitivity analysis with varying time windows into the main manuscript). This might give some understanding of the degree to which this type of confounding could have influenced their results. 

Possible selection bias: The authors state, "This database does not include deaths if no linkage is found (most likely because the decedent was never admitted to hospital)." This seems like it would tend to bias towards inclusion of individuals with a history of hospitalization, which could potentially increase the association between hospitalization and death, since individuals are also selected for inclusion by virtue of having experienced opioid-related death. If I am understanding the database correctly, this should be added as a limitation.

Minor:

Abstract: The abstract starts by stating that some hospital patients use illicit drugs while admitted, and there have been reports of patients found dead on hospital premises after using illicit drugs such as heroin. This led me to believe this was a study about use of opioids in hospitals, which is not a main focus of the analysis. I would instead start by stating why the time just after hospital discharge may be a time of heightened risk. 

Shoshana J. Herzig, MD, MPH

[LINK]

---

## [Decision Letter · Decision Letter 2]

25 Jun 2021

Dear Dr. Lewer,

Thank you very much for re-submitting your manuscript "Fatal opioid overdoses during and shortly after hospital admissions in England: case-crossover study" (PMEDICINE-D-21-01225R2) for review by PLOS Medicine.

I have discussed the paper with my colleagues and the academic editor and it was also seen again by four reviewers. I am pleased to say that provided the remaining editorial and production issues are dealt with we are planning to accept the paper for publication in the journal.

[LINK]

We look forward to receiving the revised manuscript by Jul 02 2021 11:59PM.   

Sincerely,

Beryne Odeny, 

Associate Editor 

PLOS Medicine

plosmedicine.org

Requests from Editors:

1) Thank you for providing your STROBE checklist. Please replace the page numbers with paragraph numbers per section (e.g. "Methods, paragraph 1"), since the page numbers of the final published paper may be different from the page numbers in the current manuscript.

2) If possible, please include a summary of ethnic/ racial characteristics in table 1

3) Please ensure that all weblinks are current and accessible. For example, the weblink for references #8, 9, 14, are broken

Comments from Reviewers:

Reviewer #1: I have not read the manuscript again in full, but have reviewed the authors' responses to my previous review. It is clear that they have put in substantial additional work and I learned from their detailed responses to my comments. 

My main residual concern relates to the authors' response in point 13 - I am not fully persuaded. I write death certificates and liaise with coroners when on clinical duty. In hospitalised patients, where deaths are sudden and unwitnessed, it is sometimes hard to attribute a cause of death. Where someone is found dead in the community - with no access to regular patient observations, blood test results, etc - I imagine things are even less certain. 

I am not convinced that toxicology, or a coroner reviewing what may be limited information, really helps here. Finding opiates in somebody who is opiate dependent does not tell us whether their death was a result of accidental poisoning (X42, etc), a result of the condition that recently put them into hospital, or a result of complications associated with that hospitalisation (venous thromoembolism, etc). 

I would like to see greater acknowledgement of the inherent difficulty in attributing cause of death. If data are available on the proportion of these deaths that were witnessed, and the proportion of cases that underwent post mortem, that should be presented. I would like to see mention of misattribution of cause of death as a possible alternative explanation for the observed mortality patterns in the discussion. 

Minor further point: In table 2, I struggle to see how the median can be 1, where only 40% of the observations are 1? This may be a rounding issue.

I continue to think that these results merit publication in a high impact journal, such as PLOS Medicine.

Dr Tom Yates

Imperial College London

Reviewer #2: Thanks for revised manuscript and comprehensive responses to my queries. From my perspective my original queries have been answered with the additional information and revisions to the manuscript. 

The results with the subset of ICD-10 codes (T40.1) are similar to the main results, this is reassuring that there aren't issues around the specificity of the ICD-10 codes used to identify cases. 

 The extra information on the ICD-10 codes and time-stratified analyses are helpful additions. I don't think there's evidence that time-trend exists in this data that would cause a positive bias in the outcome-exposure relationships.

Good study and very nicely presented figures.

Reviewer #3: Overall:

The revisions to the manuscript addressed all the concerns and suggestions made in the first review round. My remaining suggestions are:

- The use of the word "proximity" initially I understood it as a space measure, which is not the focus of this paper. It may be worth adding an adjective to clarify the intended meaning, i.e., "time proximity." 

- As the use of illicit, more potent, and fatal opioids (mainly fentanyl) increases in our populations, I wonder whether the estimates have a significant upward time trend (i.e., risk of death in the two weeks after hospital discharge was higher for patients admitted in 2018-2019 than for patients admitted in 2010-2011). 

Reviewer #4: The addition of the sensitivity analyses varying the control window has strengthened the manuscript.

[LINK]

---

## [Decision Letter · Decision Letter 3]

5 Aug 2021

Dear Dr Lewer, 

On behalf of my colleagues and the Academic Editor, Dr. Vikram Patel, I am pleased to inform you that we have agreed to publish your manuscript "Fatal opioid overdoses during and shortly after hospital admissions in England: case-crossover study" (PMEDICINE-D-21-01225R3) in PLOS Medicine.

PRESS

Sincerely, 

Beryne Odeny 

Associate Editor 

PLOS Medicine